# HIV Care Cascade among Prisoners of the Mandalay Central Prison in Myanmar: 2011–2018

**DOI:** 10.3390/tropicalmed5010004

**Published:** 2020-01-01

**Authors:** Nang A Mwe Nom, Khine Wut Yee Kyaw, Ajay M. V. Kumar, San Hone, Thida Thida, Thet Wai Nwe, Pyae Soan, Thurain Htun, Htun Nyunt Oo

**Affiliations:** 1National AIDS Programme, Ministry of Health and Sports, Sagaing 02371, Myanmar; 2International Union against Tuberculosis and Lung Disease, 75006 Paris, France; dr.khinewutyeekyaw2015@gmail.com (K.W.Y.K.); sathyasaakshi@gmail.com (A.M.V.K.); 3International Union against Tuberculosis and Lung Disease, Mandalay 05021, Myanmar; drthurain07@gmail.com; 4International Union against Tuberculosis and Lung Disease, South-East Asia Office, New Delhi 110016, India; 5Yenepoya Medical College, Yenepoya University, Mangaluru 575018, India; 6National AIDS Programme, Ministry of Health and Sports, Nay Pyi Taw 15011, Myanmar; sanhone@googlemail.com (S.H.); dr.pyaesoan@gmail.com (P.S.); dr.tunnyuntoo13@gmail.com (H.N.O.); 7Department of Medical Research, Ministry of Health and Sports, Pyin Oo Lwin 05085, Myanmar; drthidamph@gmail.com; 8Central Epidemiology Unit, Department of Public Health, Ministry of Health and Sports, Nay Pyi Taw 15011, Myanmar; dr.thetwainwe@gmail.com

**Keywords:** attrition, default, viral suppression, incarcerated populations, people living in closed settings

## Abstract

Prisoners have a higher HIV prevalence and higher rates of attrition from care as compared with the general population. There is no published evidence on this issue from Myanmar. We assessed (1) HIV test uptake, HIV positivity, and enrollment in care among newly admitted prisoners between 2017 and 18 (2) Treatment outcomes among HIV-positive prisoners enrolled in care between 2011 and 18. This was a cohort study involving secondary analysis of program data. Among 26,767 prisoners admitted to the Mandalay Central Prison between 2017 and 2018, 10,421 (39%) were HIV-tested, 547 (5%) were HIV-positive, and 376 (69%) were enrolled in care. Among the 1288 HIV-positive prisoners enrolled in care between 2011 and 2018, 1178 (92%) were started on antiretroviral therapy. A total of 883 (69%) were transferred out (post-release) to other health facilities, and among these, only 369 (42%) reached their destination health facilities. The final outcomes (censored on 30 June 2019) included the following: (i) Alive and in care 495 (38%), (ii) death 138 (11%), (iii) loss to follow-up 596 (46%), and (iv) transferred out after reaching the health facilities 59 (5%). We found major gaps at every step of the HIV care cascade among prisoners, both inside and outside the prison. Future research should focus on understanding the reasons for these gaps and designing appropriate interventions to fill these gaps.

## 1. Introduction

Globally, 37.9 million people were living with HIV (PLHIV) by the end of 2018 [1]. In the same year, there were 1.7 million new infections and nearly half of them occurred in key populations such as female sex workers (FSW), men who have sex with men (MSM), people who inject drugs (PWID), migrants, and people living in prisons and closed settings [1].

The global community has pledged to end the AIDS epidemic by 2030 as part of the larger sustainable development goals [2]. As an interim milestone, the UNAIDS has set the 90-90-90 targets (diagnose 90% of all PLHIV, provide antiretroviral therapy to 90% of those diagnosed, and achieve viral suppression in 90% of those treated) to be achieved by 2020 [3]. By 2018, an estimated 79% of PLHIV knew their status, 78% accessed antiretroviral therapy (ART), and 86% had suppressed viral loads [1].

Bridging the remaining gap is the most challenging part and this will not happen unless we make efforts to reach out to the most vulnerable and marginalized key populations. One such group which has faced gross neglect over the years is people living in prisons and closed settings. Every year, nearly 11 million people are incarcerated globally and this number is increasing [4,5]. Most of them are men and come from economically and socially vulnerable backgrounds, and it is estimated that 3.8% of global prison population are living with HIV [6].

The United Nations Office on Drugs and Crime (UNODC) and the World Health Organization (WHO) recommend that all prisoners be provided with a comprehensive package of HIV prevention and care services, including voluntary HIV counseling and testing and ART [7]. Unfortunately, the reality is different and there are many gaps in the HIV care cascade among prisoners. A systematic review reported that there was a high rate of loss to follow-up (LFU) when they return to the community [8]. Therefore, strengthening the continuum of care after release from prison is an integral part of reducing the spread of HIV infection in the community.

Myanmar is faced with a concentrated HIV epidemic with a prevalence of 0.57% in the general population aged 15 years and above [9], however, HIV prevalence has been higher in key population groups such as PWID (34.9%), FSW (14.6%), MSM (11.6%), and prisoners (7.7%) [9,10]. The National AIDS Program (NAP) of the Ministry of Health and Sports has implemented HIV care services in selected prisons of Myanmar since 2011, however, there has never been a systematic assessment as to how well this is being implemented.

Hence, we conducted an operational research in the Mandalay Central Prison with the following objectives: (1) To assess the number and proportion tested for HIV, found HIV-positive, and enrolled in HIV care among the prisoners who were newly admitted between 2017 and 2018 (2) Among the HIV-positive prisoners enrolled in care between 2011 and 2018 to assess, (a) the treatment outcomes, number tested for viral load and found to be virally suppressed (b) the sociodemographic and clinical factors associated with attrition (death and LFU) and not reaching the destination health facility after being transferred out (post-release).

## 2. Materials and Methods

### 2.1. Study Design

This was a cohort study involving secondary analysis of routinely collected program data.

### 2.2. Setting

Myanmar is a lower-middle income country in Southeast Asia with a population of 52 million. There are 43 prisons and 48 camps in the country with an estimated 79,668 prisoners, in 2018, which was an increase by 33% as compared with 2015 [4,10]. There are three central prisons in Myanmar, one each in Insein, Mandalay, and Tharyarwati. In 2018, a training manual containing standard operating procedures on prison health management was developed jointly by the Ministry of Health and Sports and the Ministry of Home Affairs [11].

This study was conducted at the Mandalay Central Prison. There is a 25-bed hospital, which is manned by a medical superintendent, one dental surgeon, one health assistant, five nurses, three X-ray technicians, and one compounder, within the prison. In addition, HIV care is provided by the NAP in collaboration with the staff of the International Union Against Tuberculosis and Lung Disease (The Union) who run the Integrated HIV care (IHC) program in Myanmar [12,13]. The medical doctors from The Union attend the prison hospital once a week to run the ART clinic.

#### 2.2.1. HIV Testing

Starting from February 2017, all the prisoners are expected to be routinely offered HIV testing and counseling at the time of entry. HIV testing follows the national algorithm and involves conducting a maximum of three tests. The first rapid HIV test (Alere Determine test) is done at the prison hospital. If non-reactive, no further tests are conducted. If reactive, the blood sample is sent to a NAP health facility outside the prison for confirmation. The confirmation involves testing with two more rapid HIV tests (Uni-Gold HIV and STAT PAK) and if both are reactive, the person is considered HIV-positive.

The practice of HIV testing and its documentation are varied in male and female prisons. In the male prison, HIV testing is performed four days a week (2 days for pretrial detainees and 2 days for convicts). A list of newly admitted prisoners is prepared by the prison authorities and HIV testing is offered according to that list. If someone listed is not present in the prison cell on that particular day, he is not offered HIV testing again. It is possible that a prisoner could be listed and tested twice (first time as pretrial detainee and second time as a convict). In the female prison, there is no system that prepares a separate list for HIV testing, given the relatively small number of prisoners. A responsible medical staff of the female prison offers the HIV test and documents in the records.

#### 2.2.2. Linkage to HIV Care

All convicted prisoners diagnosed as HIV-positive (including previously known HIV-positive) are enrolled in HIV care at the ART clinic within the prison hospital. ART is provided according to the prevailing national guidelines and Myanmar follows a “test and treat” strategy currently [14,15,16]. Pretrial detainees with less than six months of sentence are not enrolled in care and not initiated on ART unless they have coexisting TB, pregnancy, or clinically severe disease (WHO stage 3 and 4 or CD count <350) and this is due to concerns that pretrial detainees can be lost to follow-up from care and can develop resistance to ART. However, they receive pre-ART care including opportunistic infection management, cotrimoxazole preventive therapy, and isoniazid preventive therapy.

#### 2.2.3. Recording and Reporting

The HIV testing process is documented in the “HIV testing register” and all HIV-positive individuals are documented in the “HIV-positive register”. A unique identifier is provided to every prisoner when they enroll in HIV care and a patient file and a booklet is opened. The patient files are stored at the prison hospital and the booklet is provided to prisoners. The details of HIV care are recorded in these patient files and electronically transcribed to a database maintained by the NAP in collaboration with The Union (NAP-Union database). The database has many inbuilt checks and regularly checked for data inconsistencies and validated by The Union staff.

#### 2.2.4. Transferred Out and Post-Release Care Continuum

All prisoners receiving ART are transferred out to a health facility of their convenience at the time of release from the prison facility. A transfer form is completed and given to the patient. The transferred-out details are also recorded in the electronic database. When transferred to an IHC facility, the feedback about reaching the destination facility gets automatically recorded in the central database. If patients that are transferred to an IHC clinic do not reach the clinic, they are tracked by the Union staff to ensure continuation of care, however, for patients transferred out to non-IHC facilities, there is no system of tracking and follow-up.

### 2.3. Study Population

For objective one, all prisoners newly admitted to the Mandalay Central Prison from 1 February 2017 to 31 December 2018 were included. These included both pretrial detainees and convicts. For objective two, all the HIV-positive prisoners enrolled in care from 1 June 2011 to 31 December 2018 were included.

### 2.4. Data Variables, Sources of Data, and Data Collection

For objective one, we collected aggregate numbers of newly admitted prisoners (obtained from prison authorities), numbers tested for HIV (from HIV testing register), numbers found HIV-positive (from HIV-positive register), and numbers enrolled in care (from NAP-Union database). The data were collected for male and female prisoners separately.

For objective two, we extracted the patient-wise data from the NAP-Union database which contained key demographic and clinical variables. From this, we prepared a list of “transferred out” patients. For patients transferred out to IHC facilities, information about continued care and treatment outcomes were extracted from the NAP-Union database. For patients transferred to non-IHC facilities, we coordinated with the WHO-appointed staff of the corresponding NAP team, who searched the patient records manually at the destination health facilities and provided treatment outcome details of people who reached. These were merged into the NAP-Union database and a master dataset was created for analysis.

### 2.5. Analysis and Statistics

For objective one, we calculated proportions of HIV testing, HIV positivity, and enrollment in care and depicted in the form of a flowchart. For objective two, we used frequencies and proportions for categorical variables and mean (standard deviation) or median (interquartile range) for continuous variables.

There were four final outcomes for each patient which included: (i) alive and in care, (ii) death, (iii) LFU, and (iv) transferred out. If a patient was alive and in care as of the censor date (30 June 2019), we considered the outcome as “alive in care”. If the patient died due to any reason during care, the outcome was recorded as “death”. The LFU was recorded as the outcome for the following: (i) patients who were released from the prison without a formal transfer out process (ii) patients who were transferred out but did not reach the destination health facility, and (iii) patients who reached the destination health facility, but then interrupted treatment for a consecutive period of three months or more. If patients who reached the destination health facility were transferred to another health facility for continuing care, then the outcome was recorded as “transferred out”. Patients with viral load <1000 copies per ml were considered to be virally suppressed. The most recent viral load test results were considered for this calculation.

The time to outcome was calculated from the date of enrollment into HIV care to the final outcome date or the censor date, whichever was earlier. We created a composite outcome indicator called “attrition” by combining death and LFU. We calculated the incidence rate of attrition (and 95% confidence intervals) by dividing the number of attritions by the person-years of follow-up. We initially planned to use a Cox proportional hazards model, but the proportionality assumption was not met. Hence, we used a modified Poisson regression with robust error variance and adjusted incidence rate ratios were calculated, to assess factors associated with attrition. In addition to age and sex, variables found to be statistically significant (*p* value < 0.05) in unadjusted analysis were included in the final model. We checked for collinearity using a variance inflation factor (VIF) and variables with a VIF value >10 were excluded. Among the transferred-out patients, we also assessed factors associated with “not reaching” the destination health facilities using the same approach described above.

### 2.6. Ethics Approval

Permission to conduct the study and access the data was obtained from the NAP and the prison health department of Ministry of Home Affairs. Ethics approval was obtained from the Ethics Review Committee, Department of Medical Research, Myanmar (approval number 2019-49) and the Ethics Advisory Group of The Union, Paris (approval number 05/19). As this study involved a review of existing records, the ethics committees waived the need for individual informed consent.

## 3. Results

### 3.1. HIV Testing Uptake and Linkage to HIV Care

Among the 26,767 inmates newly admitted into the prison between 2017 and 2018, 10,421 (39%) were HIV tested and 560 (5%) were HIV-positive. Of the latter, 376 (67%) were enrolled in care. HIV testing rates were higher among males (40%) as compared to females (29%). Similarly, the rates of enrollment into care were higher among males (68%) as compared to females (57%) (Figure 1).

### 3.2. Demographic and Clinical Characteristics

There was a total of 1288 HIV-positive people who were enrolled in care at the IHC prison clinic over an eight-year period from 2011 to 2018. Of them, 1162 (90%) were male and the median (interquartile range, IQR) age was 32 (27 to 38) years. The median (IQR) CD4 count at baseline was 256 (158 to 387) cells/µL and 536 (42%) were in WHO clinical stage 3 or 4. Nearly half of them had anemia (classified as per WHO guidelines) and 876 (68%) were people who inject drugs (PWID) [17] (Table 1).

### 3.3. Treatment Outcomes after Enrollment into HIV Care

The journey of the HIV-positive people enrolled in care is depicted in Figure 2. Among 1288 patients, 1178 (91.5%) were started on ART in the IHC prison clinic, while 48 (3.7%) people died, 53 (4.1%) were transferred out, and nine (0.7%) were LFU before starting ART. Among the patients who were started on ART, 270 (22.9%) were alive in care in the prison as of the censor date, 67 (5.7%) had died, 11 (0.9%) were LFU, and 830 (70.5%) were transferred out. Thus, a total of 883 were transferred out from prison to other health facilities outside the prison. Of these, only 369 (41.7%) reached their destination health facilities, while the remaining were considered LFU. Overall, of the 1288 people enrolled, 495 (38.4%) were alive and in care as of the censor date, 138 (10.7%) had died, 596 (46.3%) were LFU, and the remaining were transferred out. A total of 805 (63%) people had a documented viral load test result and of them, 760 (94%) were found to be virally suppressed.

### 3.4. Factors Associated with Attrition

The total duration of follow-up of 1288 people enrolled was 3132 person-years (median follow-up was 2.14 years per person) and there was a total of 734 attritions (deaths and LFUs). Thus, the incidence rate of attrition was 23 (95% confidence interval 22 to 25) per 100 person-years follow-up. Nearly half of the attrition occurred during the first two years of follow-up (Figure 3). In adjusted analysis, the incidence rate of attrition was significantly higher among males and those who were illiterate (Table 2).

### 3.5. Factors Associated with Not Reaching the Destination Health Facilities

Among prisoners who were transferred out, 514 (58%) did not reach the destination health facilities. The drop out was significantly higher among males as compared with females, those who were in WHO stage 1 and 2 as compared with WHO stage 3 and 4 and those who were transferred to NAP and other NGO health facilities as compared to IHC clinics (Table 3).

BMI and mode of transmission were not included in the adjusted analysis due to collinearity with sex. Hepatitis B was not included due to collinearity with Hepatitis C.

## 4. Discussion

This is the first study from Myanmar describing the cascade of HIV care among prisoners while in prison and post-release. There were three key findings. First, HIV test uptake was low and only one in four prisoners received a HIV test. Second, enrollment into HIV care was modest at about 70%. Finally, there was high attrition (~50%), among the pot-release prisoners.

Our study had three strengths. First, the study was conducted in one of the largest prisons in Myanmar and we had a large sample size, which enabled a robust analysis and minimized the effect of random variation. Second, we used the NAP-Union patient database as the primary database of our study, which is quality-assured with inbuilt checks and periodic validation by the data managers. Third, we followed the Strengthening the Reporting of Observational Studies in Epidemiology (STROBE) guidelines for reporting the study findings [18].

This study also had some limitations. First, we had to rely on aggregate data for fulfilling our first objective because (i) there were multiple registers maintained in the prison which were non-standardized and (ii) there was no unique identifier to link these registers. Hence, we were unable to perform an individual cohort analysis. Second, the aggregate numbers included both pretrial detainees and prisoners between 2017 and 2018. It is possible that many of the pretrial detainees could have become convicts in the same study period, and thus they could have been counted and tested twice. We do not know the impact of this on the overall estimate, and therefore request the readers to interpret the HIV test uptake data with caution. Third, we could have overestimated the attrition rate because we only tracked transferred-out patients, in the destination facilities, and considered anyone not reaching their facility as LFU. It is possible that some patients could have reached health facilities other than the facility that they we transferred to and could have continued in care. It was not possible to track such patients because there is no centralized, national-level electronic patient database. Fourth, we could not systematically assess the reasons for the gaps in the cascade of care. This requires qualitative research methods and should be a topic of future research. Despite these limitations, the study has many policy and practice implications.

First, we recommend implementation of provider-initiated counseling and HIV testing with an “opt-out” approach to improve the HIV test uptake. This strategy has shown to be feasible and effective for detecting additional HIV positive cases in other studies [19,20].

Second, to improve the linkage to HIV care, we recommend that all HIV-positive prisoners (including pretrial detainees) be enrolled in care and started on ART, in line with the global and national “test and treat” strategy. The current practice of not starting ART among pretrial detainees (with less than six months of sentence) is not ethical and needs to be discontinued.

Third, a simplified, uniform recording and reporting system that will capture all the steps of HIV care in prison should be developed and implemented. We recommend that there be only one register which captures HIV test status, test results, and information on enrollment into care. From this, simple indicators (such as number of prisoners admitted, proportion HIV tested, HIV positivity, and proportion enrolled into care) can be calculated and captured in the prison monthly report.

Fourth, we recommend strengthening the counseling of prisoners at the time of release, especially those who are not literate and those in WHO stage 1 and 2, who may be clinically stable and may not fully understand the dangers of discontinuing care. The counseling package should also provide information on health facilities which provide ART care so that the prisoner can choose the health facility of their convenience, the address of the destination health facility, and contact details of the health care provider. Randomized controlled trials have reported the benefit of prerelease counseling and adherence support [21,22].

Fifth, there is an urgent need to develop a system of tracking patients who are transferred out of the prison, especially those who are transferred out to non-IHC health facilities, where the LFU was higher as compared with IHC facilities. We also suggest documenting the master patient index number in the transferred-out forms along with telephone numbers of the patients. A list of transferred-out patients should be prepared periodically and shared with the corresponding health facilities over email. This will help the NAP staff, at the destination health facilities, track and follow-up over the phone. There is strong evidence of phone-based tracking works in reducing LFU [23,24,25]. A system of bidirectional feedback mechanism between different health facilities or organizations should be developed. Finally, we recommend that NAP consider setting up a national, web-based, HIV patient database which will help in easy tracking and routine cohort analysis and monitoring.

The released prisoners face a lot of challenges when they reenter into the community which include substance abuse, stressful life experience, stigma, and discrimination in trying to fulfill the basic needs of life [26,27]. In such a situation, HIV care may get neglected. It is recommended that the prison authorities collaborate with other social support programs for a smooth transition and better reintegration into community.

In conclusion, this study described the gaps in HIV care cascade among the prisoners, both inside and outside the prison, post-release. We recommend the NAP and the Prison health department consider the practical recommendations made in the paper to improve the overall HIV care among prisoners.

## Figures and Tables

**Figure 1 tropicalmed-05-00004-f001:**
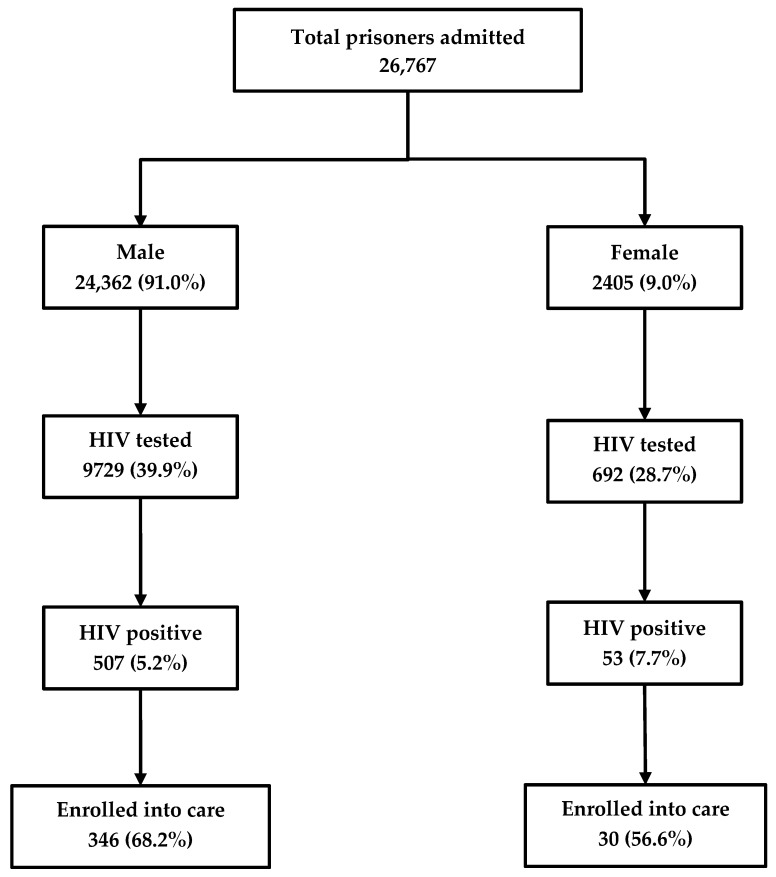
The cascade of HIV care among prisoners admitted to the Mandalay Central Prison between 2017 and 2018. HIV, human immunodeficiency virus.

**Figure 2 tropicalmed-05-00004-f002:**
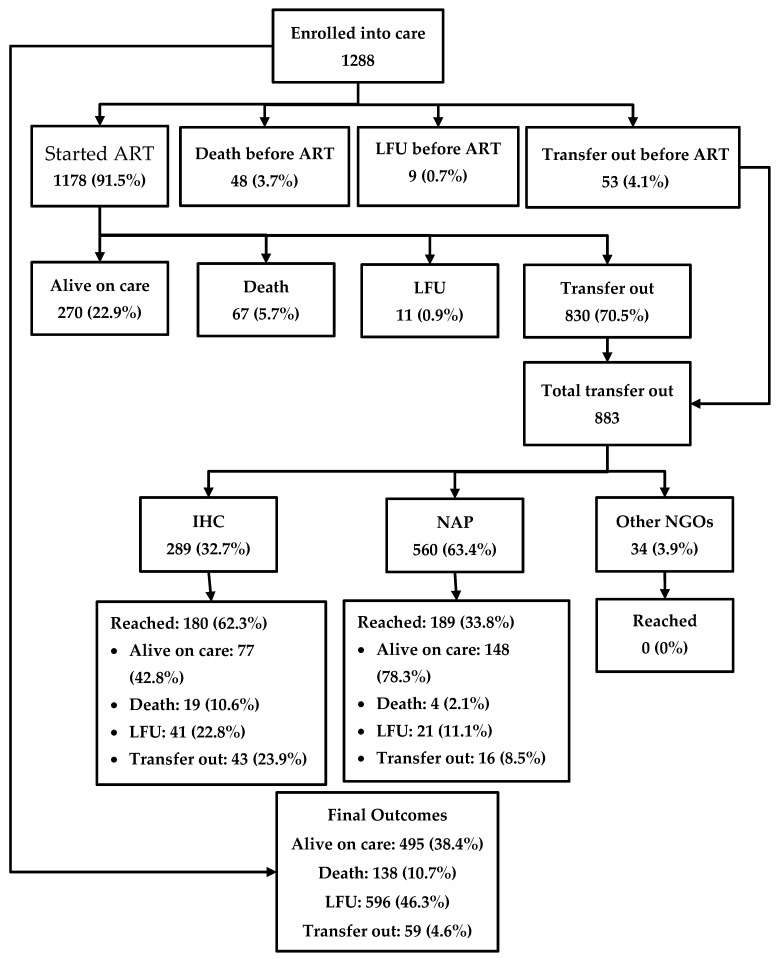
The cascade of HIV care among prisoners enrolled to HIV care at the Mandalay Central Prison between 2011 and 2018. ART, anti-retroviral therapy; NAP, National AIDS program; IHC, integrated HIV care; and LFU, lost to follow-up.

**Figure 3 tropicalmed-05-00004-f003:**
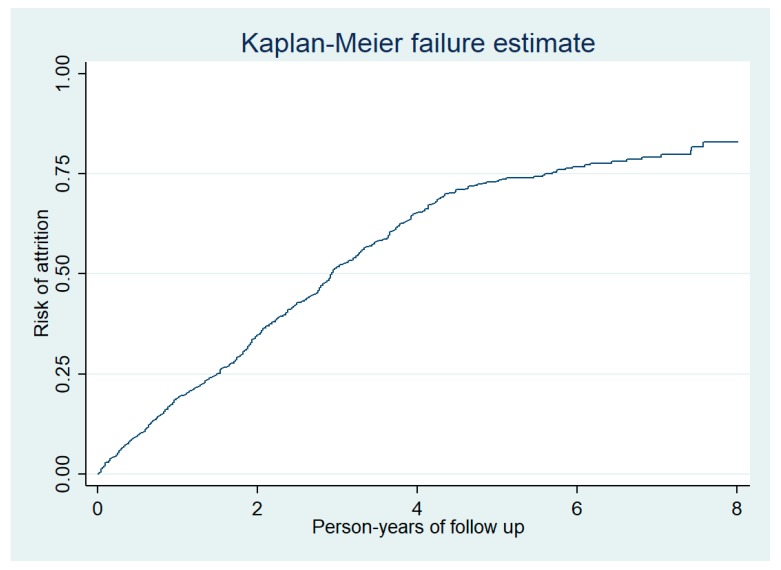
Cumulative risk of attrition among prisoners enrolled into HIV care at the Mandalay Central Prison between 2011 and 2018.

**Table 1 tropicalmed-05-00004-t001:** Sociodemographic and clinical characteristics of prisoners enrolled in the integrated HIV care (IHC) program in Mandalay central prison between 2011 and 2018.

Characteristics	N	(%)
Total	1288	(100)
Age (years)		
15–29	475	(36.9)
30–44	694	(53.9)
≥45	119	(9.2)
**Sex**		
Male	1162	(90.2)
Female	126	(9.8)
**Year of enrollment**		
2011–2013	258	(20.0)
2014–2015	314	(24.4)
2016–2018	716	(55.6)
**Mode of transmission**		
Heterosexual	328	(25.5)
PWID	876	(68.0)
Other	36	(2.8)
Unknown	48	(3.7)
**Literate**		
No	125	(9.7)
Yes	1101	(85.5)
Missing	62	(4.8)
**TB at enrollment**		
No	1138	(88.4)
Yes	147	(11.4)
Missing	3	(0.2)
**Baseline WHO staging**		
Stage 1	334	(25.9)
Stage 2	411	(31.9)
Stage 3	485	(37.7)
Stage 4	51	(4.0)
Missing	7	(0.5)
**CD4 count (cells/µL)**		
<200	437	(33.9)
200–349	421	(32.7)
350–499	252	(19.6)
≥500	140	(10.9)
Missing	38	(2.9)
**Anaemia**		
No anaemia	685	(53.2)
Mild	299	(23.2)
Moderate	216	(16.8)
Severe	46	(3.6)
Missing	42	(3.2)
**Hepatitis B**		
Negative	1105	(85.8)
Positive	122	(9.5)
Missing	61	(4.7)
**Hepatitis C**		
Negative	391	(30.4)
Positive	837	(65.0)
Missing	60	(4.6)
**BMI (kg/m^2^)**		
Underweight	190	(14.8)
Normal	903	(70.1)
Overweight or Obese	75	(5.8)
Missing	120	(9.3)
**ART Regimen**		
No ART	111	(8.6)
AZT based	41	(3.2)
D4T based	146	(11.3)
TDF based	988	(76.7)
ABC based	2	(0.2)

IHC, integrated HIV care program; WHO, World Health Organization; CD4, cluster differentiation 4; µL, microliter; TB, tuberculosis; BMI, body mass index; kg/m^2^, kilogram per meter square; PWID, people who inject drugs; ART, antiretroviral therapy; AZT, zidovudine; D4T, stavudine, TDF, tenofovir; and ABC = abacavir.

**Table 2 tropicalmed-05-00004-t002:** Sociodemographic and clinical characteristics associated with attrition among prisoners enrolled in the HIV care in Mandalay central prison between 2011 and 2018.

**Characteristics**	**PY**	**Attrition**	**IR**	**IRR**	**95% CI**	**aIRR**	**95% CI**
**Total**	3132	734	23				
**Age (years)**							
15–29	1108	266	24	ref		ref	
30–44	1734	401	23	1.03	(0.93–1.14)	1.00	(0.90–1.12)
≥45	290	67	23	1.01	(0.84–1.20)	0.93	(0.77–1.13)
**Sex**							
Male	2735	674	25	1.22	(1.01–1.47) *	1.28	(1.04–1.56) *
Female	398	60	15	ref		ref	
**Mode of transmission**							
Heterosexual	839	180	21	ref		NE	
PWID	2097	510	24	1.06	(0.95–1.19)		
Other	95	17	18	0.86	(0.60–1.23)		
Unknown	101	27	27	1.03	(0.78–1.34)		
**Literate**							
No	314	84	27	1.22	(1.07–1.4) *	1.21	(1.05–1.39) *
Yes	2664	606	23	ref		ref	
Missing	154	44	29	1.29	(1.09–1.53) *	1.35	(1.11–1.64) *
**TB at enrollment**							
No	2797	632	23	ref		ref	
Yes	327	100	31	1.22	(1.08–1.38) *	1.10	(0.95–1.27)
Missing		2					
**Baseline WHO staging**							
Stage 1 & 2	1785	395	22	ref		ref	
Stage 3 & 4	1336	335	25	1.18	(1.07–1.3) *	1.08	(0.96–1.2)
Missing		4					
**CD4 count (cells/µL)**							
<200	1094	261	24	ref		ref	
200–349	1088	222	20	0.88	(0.78–0.99) *	0.93	(0.82–1.05)
350–499	589	131	22	0.87	(0.76–1.00)	0.90	(0.78–1.05)
≥500	356	83	23	0.99	(0.85–1.16)	1.07	(0.90–1.26)
Missing	5	37	751				
**Anaemia**							
No anaemia	1701	369	22	ref		ref	
Mild	795	162	20	1.01	(0.89–1.14)	0.96	(0.84–1.09)
Moderate	512	134	26	1.15	(1.02–1.31) *	1.09	(0.95–1.26)
Severe	106	32	30	1.29	(1.05–1.58) *	1.20	(0.95–1.52)
Missing	18	37	211	1.64	(1.43–1.86) *	0.68	(0.38–1.23)
**Hepatitis B**							
Negative	2793	613	22	ref		NE	
Positive	300	63	21	0.93	(0.78–1.11)		
Missing	40	58	146	1.71	(1.59–1.85) *		
**Hepatitis C**							
Negative	1081	204	19	ref		ref	
Positive	2012	473	24	1.08	(0.97–1.21)	1.10	(0.98–1.24)
Missing	39	57	145	1.82	(1.63–2.04) *	1.68	(1.39–2.03) *
**BMI (kg/m^2^)**							
Underweight	406	121	30	1.17	(1.03–1.32) *	NE	
Normal	2307	493	21	ref			
Overweight	203	34	17	0.83	(0.64–1.07)		
Missing	216	86	40	1.31	(1.16–1.49) *		

Attrition, lost to follow-up or death; IHC, integrated HIV care program; PY, person years of follow-up; IR, incidence rate; IRR, incidence rate ratio; aIRR, adjusted incidence rate ratio; CI, confidence interval; WHO, World Health Organization; CD4, cluster differentiation 4; µL, microliter; TB, tuberculosis; BMI, body mass index; kg/m^2^, kilogram per meter square; PWID, people who inject drugs; ref, reference; *, statistically significant with *p* value < 0.05; NE, not estimated.

**Table 3 tropicalmed-05-00004-t003:** Sociodemographic and clinical characteristics associated with not reaching the destination health facility among prisoners transferred out from the Mandalay central prison between 2011 and 2018.

Characteristics	Total	Not Reached	(%)	RR	(95% CI)	aRR	(95% CI)
**Total**	883	514	(58.2)				
**Age (years)**							
15–29	338	202	(59.8)	ref		Ref	
30–44	467	275	(58.9)	0.99	(0.88–1.11)	1.01	(0.90–1.13)
≥45	78	37	(47.4)	0.79	(0.62–1.02)	0.83	(0.65–1.06)
**Sex**							
Male	809	488	(60.3)	1.72	(1.25–2.35) *	1.42	(1.04–1.92) *
Female	74	26	(35.1)	ref		Ref	
**Mode of transmission**							
Heterosexual	186	94	(50.5)	ref		NE	
PWID	637	390	(61.2)	1.21	(1.04–1.41)		
Other	23	12	(52.2)	1.03	(0.68–1.57)		
Unknown	37	18	(48.6)	0.96	(0.67–1.38)		
**Literate**							
No	89	53	(59.6)	1.02	(0.85–1.22)	1.17	(0.98–1.39)
Yes	744	436	(58.6)	ref		Ref	
Missing	50	25	(50.0)	0.85	(0.64–1.13)	1.07	(0.82–1.40)
**TB status at enrollment**							
No	773	449	(58.1)	ref		Ref	
Yes	107	63	(58.9)	1.01	(0.86–1.20)	1.16	(0.97–1.39)
Missing	3	2	(66.7)				
**Baseline WHO staging**							
Stage 1 & 2	517	323	(62.5)	1.19	(1.06–1.34) *	1.17	(1.03–1.33) *
Stage 3 & 4	363	190	(52.3)	Ref		Ref	
Missing	3	1	(33.3)				
**CD4 count (cells/µL)**							
< 200	291	154	(52.9)	ref		Ref	
200–349	299	179	(59.9)	1.13	(0.98–1.30)	1.04	(0.90–1.20)
350–499	183	107	(58.5)	1.10	(0.94–1.30)	0.98	(0.83–1.15)
≥ 500	102	68	(66.7)	1.26	(1.06–1.50) *	1.17	(0.99–1.39)
Missing	8	6	(75.0)				
**Anaemia**							
No anaemia	495	309	(62.4)	ref		Ref	
Mild	211	113	(53.6)	0.86	(0.74–0.99) *	0.89	(0.78–1.02)
Moderate	138	72	(52.2)	0.84	(0.70–0.99) *	0.94	(0.79–1.12)
Severe	27	12	(44.4)	0.71	(0.46–1.09)	0.95	(0.64–1.41)
Missing	12	8	(66.7)	1.07	(0.71–1.60)	0.62	(0.27–1.41)
**Hepatitis B**							
Negative	783	448	(57.2)	ref		NE	
Positive	76	48	(63.2)	1.1	(0.92–1.32)		
Missing	24	18	(75.0)	1.31	(1.03–1.66) *		
**Hepatitis C**							
Negative	247	127	(51.4)	ref		Ref	
Positive	613	370	(60.4)	1.17	(1.02–1.35) *	0.91	(0.80–1.05)
Missing	23	17	(73.9)	1.44	(1.10–1.89) *	1.29	(0.94–1.77)
**BMI (kg/m^2^)**							
Underweight	112	65	(58.0)	0.97	(0.82–1.15)	NE	
Normal	639	383	(59.9)	ref			
Overweight	48	20	(41.7)	0.70	(0.49–0.98)		
Missing	84	46	(54.8)	0.91	(0.74–1.12)		
**Transferred-out site**							
IHC	289	109	(37.7)	ref		Ref	
NAP	560	371	(66.3)	1.76	(1.50–2.06) *	1.72	(1.45–2.04) *
NGO	34	34	(100.0)	2.65	(2.29–3.08) *	2.67	(2.23–3.18) *

Attrition, lost to follow-up or death; IHC, integrated HIV care program; follow-up, period of follow-up; IR, incident rate; RR, relative risk; aRR, adjusted relative risk; CI, confident interval; WHO, World Health Organization; CD4, cluster differentiation 4; µL, microliter; TB, tuberculosis; BMI, body mass index; kg/m^2^, kilogram per meter square; PWID, people who inject drugs; ref, reference; *, statistically significant with *p* value < 0.05; and NE, not estimated. BMI and mode of transmission were not included in the adjusted analysis due to collinearity with sex. Hepatitis B was not included due to collinearity with hepatitis C.

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
