# Peer review of "HIV Care Cascade among Prisoners of the Mandalay Central Prison in Myanmar: 2011–2018"

_tropicalmed, 2020, doi:10.3390/tropicalmed5010004_

Round 1

Reviewer 1 Report

Thank-you for the opportunity to review this article which I believe to be of interest and well-written.

A few minor comments:

I believe the introduction is interesting but is over-long and some editing would make it shorter and punchier. I don't believe it is necessary to outline the state of play of HIV in the world currently, at least not to the extent that the authors have done. 2.2 'Setting' - in the 2nd paragraph, the word 'bedded' is incorrect and should be 'bed'. 2.2.2 'Linkage to HIV Care' - Why are pre-trial detainees with less than six month of sentence not enrolled into care? 3.1 'HIV testing uptake and linkage to HIV care' - the authors state in the discussion that testing rates are low. Could there be biases in those who are tested? For example, are sicker-looking people, or IDU, more likely to be offered testing? This is not clear. 4. 'Discussion'. Could the authors comment as to why testing uptake is so low?

Author Response

Dear reviewer,

Thanks for your kind review and suggestions and I responded to each point in the attached file.

please see the attached file

with regards,

Nang A Mwe Nom (on behalf of all authors)

Reviewer 2 Report

The authors in this manuscript study the HIV care implementation in the prison facility of the Mandalay in Myanmar. This is an important study given the lack of follow up studies of the public/government/organization implementation schemes to control HIV disease. The study is noteworthy in their study designs to encompass recent and prior outcomes of such public health initiative/s. Authors have clearly stated the potential shortcomings of their study, and highlighted their suggestions/recommendations for the public initiatives to be more effective. I do not see any major flaws.

Author Response

Dear reviewer,

Thank for your kind review and suggestions

I responded your comment in the attached file in detail 

please find the attached file

with best regards,

Nang A Mwe Nom (on behalf of all authors)
